# Thoracoscopic Resection of Lung Nodules following CT-Guided Labeling in Children and Adolescents with Solid Tumors

**DOI:** 10.3390/children10030542

**Published:** 2023-03-12

**Authors:** Steven W. Warmann, Justus Lieber, Juergen F. Schaefer, Martin Ebinger, Cristian Urla, Hans-Joachim Kirschner, Ilias Tsiflikas, Andreas Schmidt, Joerg Fuchs

**Affiliations:** 1Department of Pediatric Surgery and Pediatric Urology, University Children’s Hospital Tuebingen, 72076 Tübingen, Germany; 2Department of Diagnostic and Interventional Radiology, Section of Pediatric Radiology, University Hospital Tuebingen, 72074 Tübingen, Germany; 3Department of Pediatric Hematology and Oncology, University Children’s Hospital Tuebingen, 72076 Tübingen, Germany

**Keywords:** lung metastasis, thoracoscopy, coil wire labeling

## Abstract

Resection of lung metastases in children with solid tumors is regularly hampered by limited intraoperative detectability and relevant operative trauma of the open surgical access. The aim of this study was to analyze thoracoscopic resection of lung metastases in children following CT-guided labeling with coil wires. We retrospectively analyzed data of children and adolescents undergoing this approach at our institution between 2010 and 2022 with regard to technical aspects as well as surgical and oncological data. Within this period, we performed this procedure on 12 patients wherein we resected 18 lesions (1–5 per patient). The median age of patients was 178 months (51–265). The median duration of coil wire placement was 41 min (30–173) and the median surgery time was 53 min (11–157). No conversions were necessary and no intraoperative complications occurred. Complete microscopic resection (R0) was achieved in all labeled lesions and malignant tumor components were found in 5/12 patients. Our study shows that with a careful patient selection, thoracoscopic resection of lung metastases after coil wire labeling is a safe and reproducible procedure in children. Using this approach, lesions that are expected to have a reduced intraoperative detectability during open surgery become resectable. Patients benefit from the minimally invasive surgical access and reduced operative trauma.

## 1. Introduction

Primary and secondary pulmonary metastases occur to a varying degree in children with solid tumors. Primary metastases are present in up to 20% of cases depending on tumor etiology [1,2,3]. The occurrence of pulmonary metastases and the success of their local treatment relevantly contribute to the prognosis of patients [4,5,6].

Several challenges exist for the surgery of pulmonary metastases in children. Among the most important of these challenges is the limited detectability of the lesions caused by a texture that can be very similar to the lung tissue. Furthermore, surgical access to the thoracic cavity is usually associated with relevant surgical trauma and morbidity, which potentially includes a prolonged postoperative phase for the patients. Because of these and other challenges, the decision to surgically access lung metastases or suspicious lung nodules in children with malignant tumors is regularly difficult to evaluate for the interdisciplinary team.

A possible way to deal with the cited challenges is to preoperatively label the lung metastases and to additionally use a minimally invasive approach for resection. Different techniques have been described for this purpose [7,8,9,10]. However, their use in children has rarely been reported [11,12,13]. Coil wires represent one respective technique, and their use in adults has been regularly reported [14,15,16]. The principle of the working method allows the combination of metastasis labeling and minimally invasive resection, which is of enormous interest especially for the constellation in children [17,18]. So far, the use of coil wires has not been systematically investigated. The aim of this study was to analyze thoracoscopic resection of pulmonary nodules following CT-guided coil wire labeling in children and adolescents suffering from solid tumors.

## 2. Materials and Methods

### Patients

We retrospectively analyzed the data of patients undergoing thoracoscopic resection of pulmonary nodules following CT-guided coil wire placement at our institution between 2010 and 2022. Special emphasis of the evaluation was put on patients’ data, tumor and metastasis conditions, as well as surgical and oncological outcomes (resection status, histology, and further clinical course). All children were treated according to the respective treatment trials of the Society of Pediatric Oncology and Hematology (GPOH). Initially, identification and localization of suspicious nodules was performed with a CT-scan as foreseen in the trial protocols. Additional PET sequences were not performed because they are not part of the treatment protocols, and the role of PET has not yet been finally clarified systematically for lung metastases of pediatric solid tumors. Indications for surgery were established by the institutional and the national multidisciplinary tumor board (MDT). Patients were selected for the presented procedure if indication for metastasis resection was foreseen by the protocol and if the position of the lesions allowed minimally invasive resection but lesions were not visible on the lung surface. Coil wire placement and subsequent thoracoscopic resection were performed during the same anesthesia. The reason for surgically removing the lesions without prior biopsy was that in the selected cases a complete removal (R_0_) was possible according to the diagnostic imaging. Subsequent treatment after surgery was then performed according to histology and clinical stage.

CT-guided coil wire placement is presented in Figure 1. An adapted optimized protocol for children was used for CT scanning in order to administer the least possible dose of irradiation. Children were positioned depending on the intended position and course of the coil wire. Coil wires (Lung-Marker-System, Somatex Medical Technologies GmbH, Teltow/Germany) were placed by a pediatric interventional radiologist. The responsible pediatric surgeon was present during the intervention; positioning of the coil and the course of the adjacent wire were discussed and realized with respect to the needs during minimally invasive surgery. The technique of coil wire placement was executed according to the manufacturer’s recommendations, modified for the use in children.

The correct insertion point was identified using placement of the localization grid on the skin. After local disinfection and administration of local anesthetic in the insertion area, a skin incision was performed. The puncture needle (18G) was inserted under imaging of the target area. The needle did not pierce the nodules in order to avoid dissemination of cells. The spiral of the wire was placed next to the lesion so that during surgery the lung portion containing the nodule would be elevated for an optimal resection with as little loss of healthy lung tissue as possible. After reaching the target area, the mandril of the puncture needle was removed and the cannula with the drawn-in wire was introduced. Deployment and fixation of the spiral in the target area was achieved through insertion of the marker wire into the system in the direction from proximal to distal point. Correct placement of the coil was identified via selected imaging of the target area. The fixation keel was removed from the wire and the needle system was removed, while the wire remained in the patient. A dressing was applied under sterile conditions, with which the extracorporeal part of the wire was secured. The patients were then transferred to the operation theater.

A minimally invasive atypical lung resection is presented in Figure 2. In the operation room, patients were placed as needed for the surgical procedure. Resection was performed in single lung ventilation anesthesia. However, the single lung ventilation was started only after removal of the dressing so that the retraction of the lung would not cause coil wire dislocation due to the fixation on the skin. A three-trocar technique was used for resection. After intrathoracic identification of the wire and the entrance point into the lung parenchyma, the lung tissue was elevated by grasping either the wire or the parenchyma around its point of entrance. The elevated portion of the lung was then resected in an atypical fashion using an endo-stapler device. The wire was cut intrathoracically and the resected specimen was introduced into a retrieval bag together with the rest of the coil wire. The specimen together with the bag was retrieved through an enlarged opening of the trocar site. After ending the single lung ventilation, the complete recruitment of lung areas as well as the tightness of the resection lines were ensured via direct visualization during final thoracoscopy as well as via the waterlock technique. Thoracic drains were placed if an extensive pleural adhesiolysis had been necessary or in case of several resections on the same site. Mechanical ventilation was ended directly after surgery.

## 3. Results

### 3.1. Patients’ Data (Table 1)

Between March 2010 and September 2022, 12 children and adolescents (7 male, 5 female) underwent thoracoscopic resection of pulmonary metastases after CT-guided labeling with coil wires at our institution. The median age at operation was 178 months (51–265). Diagnoses of the patients are listed in Table 1. In six of the patients, there was suspicion of a pulmonary relapse, whereas in the other six there were primary nodules, which persisted after first line chemotherapy. In six children, removal of the lesions was for diagnostic purposes, while in the other six patients, the resection was part of the therapeutic plan.

**Table 1 children-10-00542-t001:** Patients’ data.

Patient No.	Gender	Age at Surgery [M]	Diagnosis	Primary Lesion (p) or Suspected Relapse (r)	Resection for Diagnostic (d) or Therapeutic (t) Reasons
1	f	212	Osteosarcoma	p	d
2	m	229	CCS gluteal	p	t
3	f	208	Osteosarcoma	r	t
4	f	265	HCC	r	t
5	f	105	Nephroblastoma	p	d
6	m	113	Bladder RMS	p	d
7	m	164	Osteosarcoma	r	t
8	m	186	Ewing Sarcoma	p	d
9	m	63	Nephroblastoma	p	d
10	f	170	PNET	r	d
11	m	51	Nephroblastoma	r	t
12	m	197	Ewing Sarcoma	r	t

m: male, f: female, CCS: Clear Cell Sarcoma, HCC: Hepatocellular Carcinoma, RMS: Rhabdomyosarcoma, PNET: Primitive Neuroectodermal Tumor, p: primary lung nodule, r: secondary lung nodule suspicious for relapse, d: resection was for diagnostic reasons, t: resection was for therapeutic reasons.

### 3.2. CT-Guided Labeling (Table 2)

In the 12 patients, 18 lesions (1–5 per patient) were labeled. If necessary, positioning of the coils was modified in terms of minimal distance to the pleural surface (5 mm instead of 10–15 as recommended for adults) and course of the wire. Therefore, the positioning was executed in such a way that always maintained that the direction of traction during subsequent minimally invasive resection was taken into consideration. Bilateral labeling was performed in three children. The size of labeled nodules varied from 4–11 mm. Anesthesia was started in the radiological department. The median time for coil wire placement was 41 min (30–173). Coil wire placement was successful in every targeted lesion and the intended distance of less than 10 mm between the nodule and the spiral was realized in all cases. One patient with five labeled metastases had pneumothoraces on both sides, which were immediately drained by a pediatric surgeon. In four patients, minimal pneumothoraces were detected, which required no further measures; seven patients did not have a pneumothorax. No other complications occurred during coil wire placement.

**Table 2 children-10-00542-t002:** Data of CT-guided coil wire placement.

Patient No.	No. of Labeled Lesions	Site of Labeled Lesions	Size of Labeled Lesions [mm]	Time of Coil Wire Placement [min]	Thoracic Drain
1	1	left LL	4	30	no
2	5	left LL (2), left UL (2), ML	4, 5, 5, 4, 2	173	yes
3	1	left UL	6	40	no
4	1	left LL	6	31	no
5	1	right LL	7	42	no
6	1	left LL	4	38	no
7	2	left LL, right UL	11, 3	62	no
8	1	ML	6	35	no
9	1	left LL	3	54	no
10	2	left LL, right UL	1, 4	75	no
11	1	right UL	4	60	no
12	1	right UL	4	39	no

LL: lower lobe, UL: upper lobe, ML: middle lobe.

### 3.3. Thoracoscopic Resection, Pathologic Workup, and Outcome (Table 3)

After labeling, the patients were transferred to the operation room (1 level below) immediately during the same anesthesia. The dressing was removed after positioning of the patient and the single lung ventilation was initiated. At subsequent thoracoscopy, the intrathoracic course of the wire was identified. There was no case of coil wire dislocation in any of the children. The median time for resection was 52.5 min (11–157), no conversion was necessary, and no intraoperative complications occurred. In addition to the one patient who had received thoracic drains immediately after coil wire placement, five other patients received drains after surgery. All drains were removed after 2 days.

The median volume of resected specimens as determined during pathologic workup was 5.75 mL (2.1–22.6). All specimens contained the intended lesion with complete microscopic removal (R0 resection status) in every case. Histological diagnoses are listed in Table 2. Vital malignant tumor compounds were found in 5 of the 12 patients.

In the further course, two patients had pulmonary relapses seven and eight months after surgery, respectively. These two patients and a third one with a cerebral relapse died of disease. All other patients are without evidence of disease after a median follow-up of 44.5 months (0–129).

**Table 3 children-10-00542-t003:** Data of surgery, pathological workup, and outcome.

Patient No.	Time of Surgery [min]	Volume of Resected Specimen [mL]	Histology of Resected Nodule	Thoracic Drain	Hospital Stay [d]	Pulmonary Relapse	Outcome
1	45	3.6	LN	no	4	no	NED
2	157	3, 3.1, 2.9, 4.8, 7.9	LN (3), sclerosis, AH	yes (2)	8	no	DOD
3	40	22.6	Vital metastasis	no	5	no	NED
4	78	16.3	Vital metastasis	yes	9	no	NED
5	39	3.4	Fibrosis	no	4	no	NED
6	11	7.9	LN	no	6	no	NED
7	77	6.7, 2.1	Vital metastasis (2)	no	4	yes	DOD
8	105	7.8	Vital metastasis	yes	4	yes	DOD
9	60	2.8	Granuloma	yes	5	no	NED
10	63	7.5, 7.3	Vital metastasis, fibrosis	yes (right side)	7	no	NED
11	30	7.5	Granuloma	yes	3	no	NED
12	40	7.8	LN	yes	22 *	no	NED

LN: lymph node, AH: adenomatous hyperplasia, NED: no evidence of disease, DOD: died of disease *: Hospital stay of patient No.12 was prolonged due to reasons not related to the procedure.

## 4. Discussion

Over recent years, there has been an improvement of treatment results in children with solid tumors. However, in all etiologies there are conditions that are associated with a worse prognosis. Stage IV disease (occurrence of distant metastases) has been identified as such a condition. The lungs belong to the most common sites of distant metastases in children with solid tumors. Primary lung metastases, but especially also pulmonary relapses, represent a difficult task for the treatment of children. The role of surgery for lung metastases differs relevantly within the different tumor types; nevertheless, the need to surgically remove pulmonary nodules regularly occurs in affected patients. This is furthermore important since, in specific cases, resection of lung metastases can contribute to the prevention of pulmonary irradiation, which has side effects, especially in the young age group (e.g., effects on cardiac function, breast development, and others).

Commonly, there are two possible reasons for resection of suspicious lung lesions: (a) in proven pulmonary metastases or their remainders, if they persist at the end of first line chemotherapy, and (b) in cases of newly identified lesions, when a histological clarification becomes necessary in order to determine the further management of the patients. The indication for resection of pulmonary lesions is usually established by a multidisciplinary tumor board based on the actual situation of the patients within the treatment plan according to the study protocol. Several factors have to be considered with regard to surgery with some of them imposing relevant challenges for the decision-making process. Depending on the underlying diagnosis, size, and localization of a pulmonary manifestation, it might be difficult to detect nodules intraoperatively, because their texture can be similar to the lung. This might be the case, for example, in nephroblastoma or hepatoblastoma. This aspect becomes even more relevant when taking the expected operative trauma of an open access to the thoracic cavity into consideration. Often, the malignant character of a lung lesion is not clearly predictable. A relevant number of resected pulmonary nodules are of a benign character, yet there is still the need to clarify their histology [19,20].

As a consequence of these factors, a way to reliably detect the areas of pulmonary lesions during surgery as well as establishing a less traumatic method for resection has become desirable. Preoperative labeling combined with minimally invasive resection of metastases has been developed during recent years to meet these challenges. Different techniques pursuing this approach have been described, some of them also in children [7,8,9,10,11,12,13,21,22,23]. These methods include labeling with Indocyanine Green (ICG) or other fluorescending agents, radionuclear labeling, or the use of hook wires. Fluorescing agents or radio-labeling for intraoperative detection of metastases have been used for some time now. However, certain limitations remain associated with these techniques. Size and depth of the nodules within the surrounding parenchyma as well as the amount of uptake by the tissue determine the intraoperative detectability. In this regard, fluorescence and radio-labeling produce the best results if the labeled lesions are of a certain size and proximity to the lung surface. Furthermore, such substances need IV application with the risk of radiation or allergic reactions; indocyanine-green is not absorbed by all different entities and therefore limited in its use to specific tumor types, mainly liver tumors. The presented coil wire approach has thus several advantages over the other methods, because it is also feasible in very small lesions (because it is sufficient to place the coil in close proximity to the metastasis without the need for perforation or hooking it into the lesion), and it does not depend on any biological or histological behaviors of the lesions.

Minimally invasive resection of lung metastases after coil wire labeling was performed during the observation period under careful selection of patients. This selection was based on the criteria of metastasis surgery depending on tumor histology and biology as well as localization and number of lesions and finally conditions within the treatment according to the respective therapy protocol. Special consideration was also given to the fact that special challenges of surgery of metastases exist. It is well known that from a number of six–eight metastases, more lesions are regularly found intraoperatively than were described in the preoperative imaging. Furthermore, depending on the etiology and size, palpation of metastases is more difficult in the context of open surgery. Finally, the number of non-malignant foci after resection of suspicious lung findings is also known to be relevant. In summary, we performed the presented procedure in children in whom the indication for metastasectomy arose according to protocol in the course of diagnosis or therapy, and one or more of the following additional aspects were present: The need for the removal of individual nodules, the location of nodules in the parenchyma away from the surface with expected difficulty in finding them during open surgery, and the feasibility of minimally invasive resection.

An important advantage of wire labeling lies in the technical execution of resections using this approach. With this technique, it is not necessary to directly visualize the lesion during surgery. After CT-guided positioning next to the lesion, the ending of the wire delineates the area where the intended goal of resection is. Atypical lung resection including this area enables the surgeon to then safely resect the nodule.

A similar technique compared to the one we used is the labeling with hook wires. Although based on the same principle, the hook at the end of the wire seems to be associated with a higher risk of dislodgement [9]. This fact seems to be caused by the way the coils are anchored within the lung parenchyma. Putting traction on the wire for exposing the lung area intended for resection seems safer, possibly with the use of a coil for anchoring within the parenchyma. The spiral should be placed with a sufficient distance to the lung surface in order to avoid dislocation during thoracoscopic resection. However, in individual cases of our series, the distance to the surface was below 1 cm. In our experience, this is acceptable and does not preclude a successful maneuver, provided that there is good communication between the executing subspecialties.

We believe that a careful patient selection is crucial for the success of the presented approach. We performed coil wire labeling and subsequent thoracoscopic resection only if coils could be safely positioned in proximity to the intended nodules without putting vital structures at risk. Furthermore, the course of the wire is essential because it determines the way the operating surgeon is able to elevate the intended lung area for a safe and minimal resection.

In our experience, it is therefore essential that radiologists and surgeons intensively discuss the cases beforehand. Furthermore, it is important for the surgeon to be present during the radiological intervention. The position of the labeling spiral in relation to the lesion determines how the surgeon can approach the resection later on. This has implications not only for the positioning of the patient on the OR table but also for choosing the sites of the trocars. Furthermore, the direction of applying traction on the wire and lung parenchyma relevantly contributes to a complete resection of the metastasis and clear resection margins. In our study, the operating surgeon was present during CT-guided coil placement and therefore knew the direction of the wire and depth of the coil within the lung parenchyma. During thoracoscopy, the entry point of the wire into the lung was clearly identifiable. Furthermore, the lung section, in which the coil was positioned, was elevated during surgery, allowing for safe stapling across without risk of cutting the wire. Finally, being present during the labeling process enables the surgeon to immediately put a drainage in case of a relevant pneumothorax, as was the case in one of our patients.

The close cooperation of radiologists and surgeons as well as a careful patient selection contributed to the success of the reported technique. In our cohort, all intended nodules were in the resected specimens, all resections were performed with clean resection margins, and no pulmonary leakage occurred. Furthermore, the reported approach helped realize surgical resections in an organ-sparing fashion, which is always the goal in these conditions. Additionally, fast recoveries with short hospital stays were realized and the gap to potentially necessary further treatment measures was as short as possible. A limitation of this approach is present when lesions are located too peripherally. In such a case, the coils do not have enough space for a stable anchoring, which potentially results in a higher risk of dislocation. For such lesions, an alternative surgical approach is necessary.

Another limit of our study is the relatively low number of patients. However, considering the low incidence of lung metastases requiring surgery and fulfilling the inclusion criteria in children, we believe that we can present a relevant and representative patent cohort.

## 5. Conclusions

CT-guided coil wire labeling followed by thoracoscopic resection represents a safe, reliable, and minimally invasive method to resect pulmonary nodules in children with solid tumors. A careful patient selection as well as a close cooperation between the executing subspecialties is essential for the success of this method. With this approach, the challenges of lung lesions, which are likely to be unpalpable during open surgery and whose resection is associated with high surgical trauma, become less precluding for an operative approach.

## Figures and Tables

**Figure 1 children-10-00542-f001:**
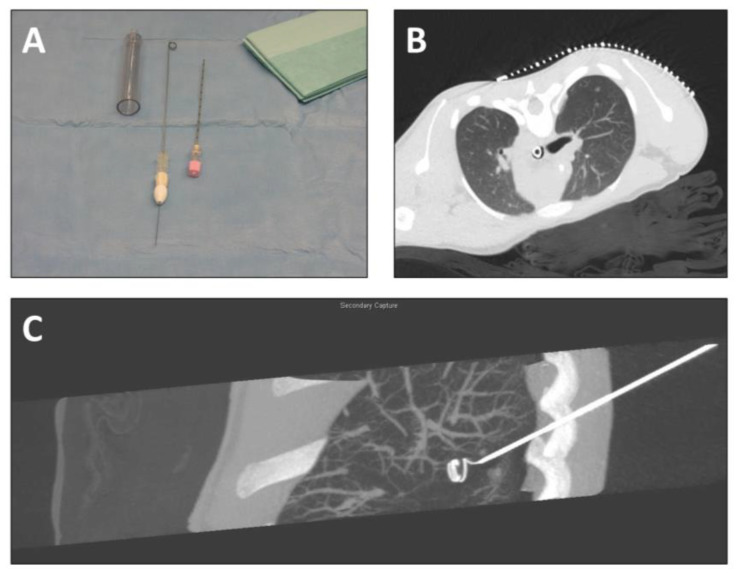
CT-guided labeling of lung nodules before thoracoscopic resection. (**A**) Equipment for labeling (introducing cannula with drawn-in wire, puncture needle); (**B**) localizing CT-scan (axial plane) with localization grid placed on the thorax above the intrathoracic nodule; (**C**) reconstruction of CT scan (sagittal plane) after coil wire placement.

**Figure 2 children-10-00542-f002:**
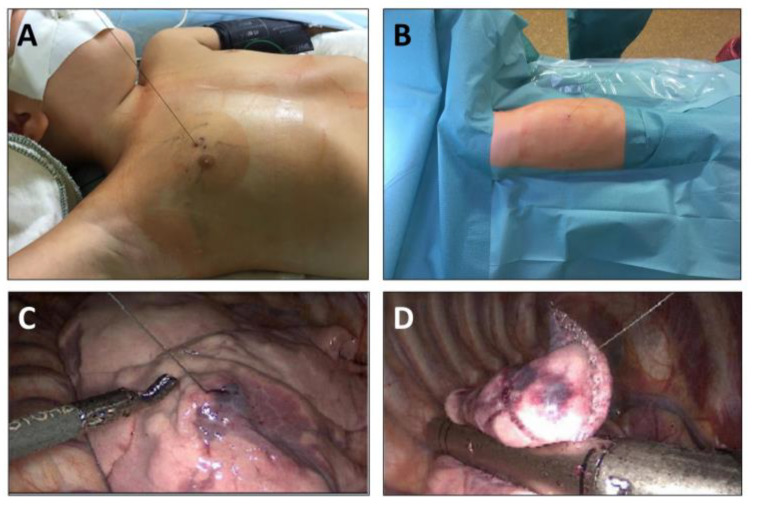
Thoracoscopic resection of lung nodules following CT-guided labeling. (**A**) Intraoperative aspect after removal of the dressing (left-sided upper-lobe nodule); (**B**) intraoperative aspect after positioning and preparation of the patient; (**C**) intraoperative view during thoracoscopy: localization of wire entering the lung parenchyma; (**D**) atypical lung resection of the labeled area using the endo-stapler.

## Data Availability

The data presented in this study are available on request from the corresponding author. The data are not publicly available due to restrictions set by the responsible ethical authority.

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
