# Peer review of "Thoracoscopic Resection of Lung Nodules following CT-Guided Labeling in Children and Adolescents with Solid Tumors"

_children, 2023, doi:10.3390/children10030542_

Round 1

Reviewer 1 Report

This is an interesting article describing the experience of a single centre in the resection of pulmonary metastases in children after CT-guided harpoon placement.

I am struck by several issues regarding the manuscript.

- The introduction is very long and should be shortened by providing a less lengthy background to the subject.

- No reference is made to indocyanine green, which is the latest method of identifying lung lesions in children, with several advantages over the method described by the authors, as it does not require such prolonged anaesthesia or radiation of the patient.

The authors describe 12 patients, but none of them has hepatoblastoma as a primary tumour, when it is one of the most frequent childhood tumours and therefore with many pulmonary metastases. The authors should explain this.

- When they state that the patients receive a single anaesthesia, do they mean that they are transferred under anaesthesia from the radiology room where the CT scan is performed to the operating theatre where they will undergo surgery? What is this transfer like, is it necessary to change floors? Is there a risk of mobilisation of the harpoon?

In bilateral metastases it is very complicated to carry out what the authors propose, as bilateral marking with a harpoon makes it difficult to carry out the subsequent bilateral thoracoscopy, due to the risk of mobilisation of the harpoons, which causes the pulmonary lesions to be left behind and identified. They should comment on this in the Discussion.

They should specify the cases in which they have used this method and why, as I assume that in 12 years they have operated on more than 12 patients with pulmonary nodules, and if this is not the case, they should not be operating on such patients in an institution that operates on one such patient per year.

Author Response

Reviewer 1

This is an interesting article describing the experience of a single centre in the resection of pulmonary metastases in children after CT-guided harpoon placement.

I am struck by several issues regarding the manuscript.

- The introduction is very long and should be shortened by providing a less lengthy background to the subject.

Authors’reply: introduction has been shortened.

- No reference is made to indocyanine green, which is the latest method of identifying lung lesions in children, with several advantages over the method described by the authors, as it does not require such prolonged anaesthesia or radiation of the patient.

Authors’ reply: This is not correct, since a reference for ICG (Ref. No. 11) has already been included.

- The authors describe 12 patients, but none of them has hepatoblastoma as a primary tumour, when it is one of the most frequent childhood tumours and therefore with many pulmonary metastases. The authors should explain this.

Authors’ reply: This study includes a selected subset of patients. Only patients which a) needed surgery for lung mets and b) in which these mets were not accessible otherwise underwent this approach. Also, hepatoblastoma lung mets usually respond and often completely regress under neoadjuvant and adjuvant chemotherapy. Therefore, the need to resect these mets is not very common.

- When they state that the patients receive a single anaesthesia, do they mean that they are transferred under anaesthesia from the radiology room where the CT scan is performed to the operating theatre where they will undergo surgery? What is this transfer like, is it necessary to change floors? Is there a risk of mobilisation of the harpoon?

Authors’ reply: This is correct. Anaesthesia is initiated in the radiological department; patients are transferred after labeling to the operating room one level below. A respective sentence has been added to the results section (results, paragraph 3.3, first line). No coil dislocation occurred during the transfer. This fact has already been stated in the results section.

- In bilateral metastases it is very complicated to carry out what the authors propose, as bilateral marking with a harpoon makes it difficult to carry out the subsequent bilateral thoracoscopy, due to the risk of mobilisation of the harpoons, which causes the pulmonary lesions to be left behind and identified. They should comment on this in the Discussion.

Authors’ reply: In contrast to the reviewer’s statement, the used device is not a harpoon wire but rather a coil wire. The coils are anchored more solidly within the lung parenchyma compared to the previously described harpoons (hook wires). A respective sentence has been added in the discussion (discussion, page 7, 4th paragraph, line 3-4).

- They should specify the cases in which they have used this method and why, as I assume that in 12 years they have operated on more than 12 patients with pulmonary nodules, and if this is not the case, they should not be operating on such patients in an institution that operates on one such patient per year.

Authors’ reply: Selection criteria for the presented approach are explained in more details in the methods section (materials and methods, paragraph 1, line 8-10).

Reviewer 2 Report

I read with interest the article and it was a pleasure to perform its review. It is a good paper and the results fully deserve to be published. My suggestions are as follows:

-          Introduction: I suggest to present briefly all methods available for marking the lung nodules

-          Row 158 misspell “”labelled””

-          Matherial and Methods

-          How did you plan to avoid the stapling across and cutting the wire, leaving the nodule with the coil behind?

-          Discussions:

-          please discuss the alternative of diagnostic before resection – why not punctioning CT-guided the nodule before resecting it?

-          Was PET-CT performed?

-          Please discuss the possibility of uniportal access, which, at least in my opinion, would allow direct palpation of some nodules.

-          Were there any other methods applied in the hospital for lung metastasectomy? How were the patients directed to one method or another?

-          Selection of the peripheral nodules should be discussed, presenting the impossibility of central placement of the coil wire and wedge-resecting this central nodules

-          Additional discussions are needed related to the advantages and disadvantages of this method and each other – eg why is it better comparing to laser resection? Or comparing to indocyanine gray iv.

-          The limits of the study must be presented.

-          According to recommendations above, at least 10-15 references need to be added.

Congratulations for the authors.

Author Response

Reviewer 2

I read with interest the article and it was a pleasure to perform its review. It is a good paper and the results fully deserve to be published. My suggestions are as follows:

- Introduction: I suggest to present briefly all methods available for marking the lung nodules

Authors’ reply: The authors thank the reviewer for this suggestion. The different methods of labeling are already mentioned in the discussion. Furthermore, reviewer 1 asked to shorten the introduction. Therefore, the authors would prefer to leave this aspect as it is.

- Row 158 misspell “”labelled””

Authors’ reply: corrected.

Matherial and Methods

- How did you plan to avoid the stapling across and cutting the wire, leaving the nodule with the coil behind?

Authors’ reply: The operating surgeon was present during CT-guided coil placement and therefore knew direction of the wire and depth of the coil within the lung parenchyma. During thoracoscopy, the entry point of the wire into the lung was clearly identifiable. Furthermore, the lung section, in which the coil was positioned, was elevated during surgery, allowing to safely stapling across without risk of cutting the wire. This aspect has been added to the discussion (discussion, page 7, last paragraph,line 8-12).

Discussions:

- Please discuss the alternative of diagnostic before resection – why not punctioning CT-guided the nodule before resecting it?

Authors’ reply: Positioning of the coils was carried out according to the manufacturer’s protocol. In this protocol, coil placement is recommended in close proximity to the lesion. Puncture of the lesion is therefore not necessary. We described in the methods section that we performed the procedure according to the protocol. In our view puncture of the lesions is not an alternative and therefore does not need to be discussed additionally.

-          Was PET-CT performed?

Authors’ reply: PET-CT was not performed.

- Please discuss the possibility of uniportal access, which, at least in my opinion, would allow direct palpation of some nodules.

Authors’ reply: Here the question of patients selection is important. The selection criteria for the presented approach have been added to the methods section (materials and methods, paragraph 1, line 8-10, see also our respective reply to reviewer 1). The two main advantages of the presented approach are a) the reduced operative trauma and b) avoidance of the need to palpate the lesions. Both aspects would be jeopardized by using a uniport access as suggested by the reviewer. The mentioned advantages have already been included in the discussion section.

- Were there any other methods applied in the hospital for lung metastasectomy? How were the patients directed to one method or another?

Authors’ reply: selection criteria were added in a clearer fashion (materials and methods, paragraph 1, line 8-10, see also our respective reply to reviewer 1).

- Selection of the peripheral nodules should be discussed, presenting the impossibility of central placement of the coil wire and wedge-resecting this central nodules

Authors’ reply: a respective section has been added in the discussion (discussion, page 8, first paragraph, line 8-10).

- Additional discussions are needed related to the advantages and disadvantages of this method and each other – eg why is it better comparing to laser resection? Or comparing to indocyanine gray iv.

Authors’ reply: differences to other approaches including ICG already have been discussed in the submitted version of the paper. The authors added another aspect in this regard (discussion, page 7, 2nd paragraph, line 11-14).  Laser resection is not an alternative for transection of the lung parenchyma in our view.

- The limits of the study must be presented.

Authors’ reply: Limits of the study have been added (discussion, page 8, 2nd paragraph).

- According to recommendations above, at least 10-15 references need to be added.

Authors’ reply: additional references have been added.

Congratulations for the authors.

Reviewer 3 Report

The authors introduced their experience regarding thoracoscopic resection of pulmonary nodules following CT-guided coil wire labeling in patients, aged from 4 years 3 months through 22 years 1 month (median age, 14 years 10 months), suffering from solid tumors. My concerns are as follows.

1.       It is noted in Materials and Methods “Special emphasis of the evaluation was put on patients’ data, tumor and metastasis conditions, as well as surgical and oncological outcome (resection status, histology, further clinical course). Unfortunately, the advantages and benefits of their new procedure are not clearly understandable, because the data were not compared with that of patients without this procedure.

2.       Four patients, Patients No 1-4, were at the age over 17 years and could not be classified as children. Therefore, some of “children” in the title and the text are better to be replaced by “children and adolescents”.

3.       The size of resected specimens are shown not with their weight after resection, but their volume during pathologic workup. I wonder why the authors did not measure the weight of resected specimens just during surgery and how they measured each volume.

4.       It is noted in Results “Vital malignant tumor compounds were found in 5 of the 12 patients”, whereas it is in Abstract “malignant tumor components were found in 7/15 resected specimens”. It seems that there is a discrepancy between the “7/15” and data shown in Table 3.

5.       As outcomes, it is noted “2 patients had pulmonary relapses 7 and 8 months after surgery, respectively. These two patients and a third one with a cerebral relapse died of disease”. I wonder whether the outcomes of lung metastases of solid tumors improved after the introduction of thoracoscopic resection following CT-guided labeling with coil wires. Although I agree with the authors that CT-guided coil wire labeling followed by thoracoscopic resection could be a safe, reliable and minimally invasive method to resect pulmonary nodules in children with solid tumors. However, I recommend the authors to show more concrete evidence to support their consideration. I personally consider that such a manuscript reporting a sort of novel surgical procedure is better to be submitted to journals in the field of thoracic or minimum invasive surgeries.

Author Response

Reviewer 3

The authors introduced their experience regarding thoracoscopic resection of pulmonary nodules following CT-guided coil wire labeling in patients, aged from 4 years 3 months through 22 years 1 month (median age, 14 years 10 months), suffering from solid tumors. My concerns are as follows.

  1. It is noted in Materials and Methods “Special emphasis of the evaluation was put on patients’ data, tumor and metastasis conditions, as well as surgical and oncological outcome (resection status, histology, further clinical course). Unfortunately, the advantages and benefits of their new procedure are not clearly understandable, because the data were not compared with that of patients without this procedure.

Authors reply: Taken together, in our view the advantages of this procedure are the combination of labeling lung metastases with a lower risk of wire dislocation and resecting them minimally-invasively. Patients benefit from a safe procedure with a low operative trauma.

  1. Four patients, Patients No 1-4, were at the age over 17 years and could not be classified as children. Therefore, some of “children” in the title and the text are better to be replaced by “children and adolescents”.

Authors’ reply: terms have been replaced.

  1. The size of resected specimens are shown not with their weight after resection, but their volume during pathologic workup. I wonder why the authors did not measure the weight of resected specimens just during surgery and how they measured each volume.

Authors’ reply: weighing of the resected specimen was part of the standardized pathological workup. Weights were taken from the pathological reports.

  1. It is noted in Results “Vital malignant tumor compounds were found in 5 of the 12 patients”, whereas it is in Abstract “malignant tumor components were found in 7/15 resected specimens”. It seems that there is a discrepancy between the “7/15” and data shown in Table 3.

Authors’ reply: the discrepancy has been noted correctly. The authors corrected the respective number in the abstract to 5/12 patients. There were 18 resected specimens in the 12 patients.

  1. As outcomes, it is noted “2 patients had pulmonary relapses 7 and 8 months after surgery, respectively. These two patients and a third one with a cerebral relapse died of disease”. I wonder whether the outcomes of lung metastases of solid tumors improved after the introduction of thoracoscopic resection following CT-guided labeling with coil wires. Although I agree with the authors that CT-guided coil wire labeling followed by thoracoscopic resection could be a safe, reliable and minimally invasive method to resect pulmonary nodules in children with solid tumors. However, I recommend the authors to show more concrete evidence to support their consideration. I personally consider that such a manuscript reporting a sort of novel surgical procedure is better to be submitted to journals in the field of thoracic or minimum invasive surgeries.

Authors’ reply: the authors did not intend to suggest that this technical approach is associated with a prognosis of patients. Therefore, we cannot follow the reviewer’s comment in this regard. We repeatedly mentioned that with this study we present a selected patient cohort. Other patients at our institution underwent different types of lung metastasis resection during the same period. Overall we experience comparable results as reported by other authors for treatment of children with stage IV tumors. The section title of the journal, for which we submitted this manuscript, is “Current Development of Pediatric Minimally Invasive Surgery”. We believe that our work fits more than sufficiently in this regard.

Round 2

Reviewer 1 Report

The authors have responded loosely to the reviewers' suggestions. They present a small series of cases of pulmonary metastases in children (12 cases in 12 years...), in which curiously no hepatoblastomas are found (they refer them to a more experienced centre?).

I recommend that the authors inform themselves about indocyanine green technology. Allergic reactions are absolutely infrequent, and have not been described in its use in pulmonary metastases. 

If only hepatoblastoma metastases were sensitive to chemotherapy and did not require surgery. The reality is different. The use of NON-INVASIVE AND NON-AGRESSIVE markers such as indocyanine green is a reality today, and I invite the authors to dive into the literature. Here are a couple of illustrative examples:

Yoshida M, Tanaka M, Kitagawa N, Nozawa K, Shinkai M, Goto H, Tanaka Y. Clinicopathological study of surgery for pulmonary metastases of hepatoblastoma with indocyanine green fluorescent imaging. Pediatr Blood Cancer. 2022 Jul;69(7):e29488.

Kitagawa N, Shinkai M, Mochizuki K, Usui H, Miyagi H, Nakamura K, Tanaka M, Tanaka Y, Kusano M, Ohtsubo S. Navigation using indocyanine green fluorescence imaging for hepatoblastoma pulmonary metastases surgery. Pediatr Surg Int. 2015 Apr;31(4):407-11.

Cho YJ, Namgoong JM, Kwon HH, Kwon YJ, Kim DY, Kim SC. The Advantages of Indocyanine Green Fluorescence Imaging in Detecting and Treating Pediatric Hepatoblastoma: A Preliminary Experience. Front Pediatr. 2021 Feb 26;9:635394.

I still do not see the proposed advantages of the technique described by the authors. It has limitations due to the size of the metastases, as it requires an adequate size for the coil to hook into it, and also requires longer anaesthesia, with transfer of the intubated patient from the radiology suite to the operating theatre. Finally, in bilateral cases where bilateral thoracoscopy is required, patient positioning involves mobilisation of the coil which distorts the location of the lesion. 

Another aspect that is not clarified is that of multiple pulmonary metastases, which is a frequent finding in paediatric tumours. In these cases they put several wires like a hedgehog?

As a final thought, the authors point out that "the use of coil wires has not been systematically investigated in children". Perhaps it is worth asking why.

Author Response

  1. The authors have responded loosely to the reviewers' suggestions. They present a small series of cases of pulmonary metastases in children (12 cases in 12 years...), in which curiously no hepatoblastomas are found (they refer them to a more experienced centre?).

Authors’ reply: The authors regret that the reviewer perceives the submitted answers as being given "loosely". In our responses, we have addressed and answered each aspect mentioned. Thus, the reason why there is no patient with hepatoblastoma in the presented collective has also been explained. We are a national surgical reference center for, among other malignancies, the treatment of liver tumors and lung metastases in childhood. The expressed assumption that we would send the children to other centers is therefore not true.

  1. I recommend that the authors inform themselves about indocyanine green technology. Allergic reactions are absolutely infrequent, and have not been described in its use in pulmonary metastases. 

If only hepatoblastoma metastases were sensitive to chemotherapy and did not require surgery. The reality is different. The use of NON-INVASIVE AND NON-AGRESSIVE markers such as indocyanine green is a reality today, and I invite the authors to dive into the literature. Here are a couple of illustrative examples:

Yoshida M, Tanaka M, Kitagawa N, Nozawa K, Shinkai M, Goto H, Tanaka Y. Clinicopathological study of surgery for pulmonary metastases of hepatoblastoma with indocyanine green fluorescent imaging. Pediatr Blood Cancer. 2022 Jul;69(7):e29488.

Kitagawa N, Shinkai M, Mochizuki K, Usui H, Miyagi H, Nakamura K, Tanaka M, Tanaka Y, Kusano M, Ohtsubo S. Navigation using indocyanine green fluorescence imaging for hepatoblastoma pulmonary metastases surgery. Pediatr Surg Int. 2015 Apr;31(4):407-11.

Cho YJ, Namgoong JM, Kwon HH, Kwon YJ, Kim DY, Kim SC. The Advantages of Indocyanine Green Fluorescence Imaging in Detecting and Treating Pediatric Hepatoblastoma: A Preliminary Experience. Front Pediatr. 2021 Feb 26;9:635394.

Authors’ reply: As a national and international surgical reference center, we are intensively involved in the various options for the treatment of childhood solid tumors. Representatives of our institution are active members in the steering committees of national and international trial groups and have been instrumental in the development of the surgical guidelines for childhood liver tumors (SIOPEL/PHITT protocol), renal tumors (SIOP/RTSG Umbrella protocol) as well as neuroblastoma and soft tissue sarcoma. In our opinion, the comment that the authors should inform themselves about the use of ICG seems tendentious and out of place at this point. ICG is used by several groups in surgery of pediatric lung metastases, although systematic experience exists only in its use for hepatoblastomas. However, ICG is not commonly used by all groups that are operating on hepatoblastoma and respective metastases on a high level. This is explained in the discussion and underlined by corresponding citations in the reference list of the manuscript (Discussion, page 7, 2nd paragraph, references 21-23). In our response to the previous review, we pointed out that the number of children with hepatoblastoma who require surgery for lung metastases is rather small compared with other entities. We did not say that this number is zero, as the reviewer implies. The reviewer’s recommendation to “…dive into the literature…” seems therefore rather curious and surprising. In fact, one of the references that the reviewer recommends the authors to study is already part of the reference list of the paper (Reference No. 21, Yoshida et al.).

  1. I still do not see the proposed advantages of the technique described by the authors. It has limitations due to the size of the metastases, as it requires an adequate size for the coil to hook into it, and also requires longer anaesthesia, with transfer of the intubated patient from the radiology suite to the operating theatre. Finally, in bilateral cases where bilateral thoracoscopy is required, patient positioning involves mobilisation of the coil which distorts the location of the lesion. 

Authors’ reply: As described in the paper, the coils do not need to perforate the lesion and to hook into the metastasis. Rather, the coils are positioned in close proximity to the metastasis, and subsequently resected together with the metastasis thoracoscopically. Therefore, the lesions do not need to have a certain size. On the contrary, it is a specific advantage of this method over for example ICG that even small lesions, that are localized deeper within the parenchyma, can be safely resected using the presented approach. A respective sentence has been added to the discussion section in order to make this aspect even clearer (page 7, 2nd paragraph, line 15-19).

  1. Another aspect that is not clarified is that of multiple pulmonary metastases, which is a frequent finding in paediatric tumours. In these cases they put several wires like a hedgehog?

Authors’ reply: As can be seen from the clinical data (Methods section and Tables 2, 3), in some patients multiple coil wires were placed and the corresponding metastases resected. As mentioned in the results, dislocation of the wires did not occur in any patient, regardless of the number of lesions marked. (Again, the authors are surprised by the somewhat inappropriate wording used by the reviewer at this point.)

  1. As a final thought, the authors point out that "the use of coil wires has not been systematically investigated in children". Perhaps it is worth asking why.

Authors’ reply: With regard to the aspects already mentioned above, the authors have no further comment on this.

Reviewer 2 Report

Round 2:

The different methods of labelling must be presented, shortly, with highlighting the author’s method and its advantages – why did you choose this method? I understand the authors concern about too long Introduction, I suggest to present them in Discussions.

Please briefly explain, in the text, why the PET-CT was not necessary.

Please briefly explain, in the text, why you chose resection without biopsy and not the biopsy + chemo/immuno.

With these info added in the text for our readers, I consider the article being complete. Congratulations for the authors!

Author Response

  1. The different methods of labelling must be presented, shortly, with highlighting the author’s method and its advantages – why did you choose this method? I understand the authors concern about too long Introduction, I suggest to present them in Discussions.

Authors’ reply: Further methods are mentioned in the discussion section (page 7, 2nd paragraph, line 6-7). The highlights and advantages of our method are now being explained in more detail (page 7, 2np paragraph, line 15-19).

  1. Please briefly explain, in the text, why the PET-CT was not necessary.

Authors’ reply: Treatment of all patients was performed according to the respective multicenter trials. Initially, identification and localization of suspicious nodules was performed by CT-scan as foreseen in the trial protocols. Additional PET sequences were not performed because they are not part of the treatment protocols, and the role of PET has not yet been finally clarified systematically for lung metastases of pediatric solid tumors. This aspect is now being explained in the text (page 2, Materials and Methods, 1st paragraph, line 6-10).

  1. Please briefly explain, in the text, why you chose resection without biopsy and not the biopsy + chemo/immuno.

Authors’ reply: The reason for surgically removing the lesions without prior biopsy were that in the selected cases a complete removal (R0) was possible according to the diagnostic imaging. The subsequent treatment after surgery was then performed according to histology and clinical stage. This is now being explained in the text (page 2, Materials and Methods, 1st paragraph, line 15-18)

With these info added in the text for our readers, I consider the article being complete. Congratulations for the authors!

Authors’ reply: The authors thank the reviewer for his constructive and quality improving comments

Reviewer 3 Report

I appreciate the authors’ effort to have upgraded their manuscript reporting their experience of CT-guided coil wire labeling for thoracoscopic resection of lung nodules in children and adolescents based on the reviewers’ comments. Minor revision on my comments below would be appreciated.

The authors described in Abstract “Our study shows that with a careful patient selection, thoracoscopic resection of lung metastases after coil wire labeling is a safe and reproducible procedure in children(probably “and adolescents”). It is described in Materials and Methods “Patients were selected for the presented procedure if indication for metastasis resection was foreseen by the protocol and if position of the lesions allowed minimally invasive resection but lesions were not visible on the lung surface”. The authors described in their response to reviewers “Other patients at our institution underwent different types of lung metastasis resection during the same period”.

I wonder what had been the indication for CT-guided coil wire labeling in the 13 years between 2010 and 2022 in their institutes, which is not described precisely and clearly in the text. I also wonder how they had managed such resections of lung nodules in children and adolescents, who had not been selected for CT-guided coil wire labeling before thoracoscopic resection, thoracotomy or thoracoscopic resections following fluorescing agents or radio-labeling for intraoperative detection of metastases.

I believe that concrete and specified descriptions regarding “a careful patient selection” for the CT-guided coil wire labeling in Abstract and Discussion would be informative and helpful for readers.

Author Response

I appreciate the authors’ effort to have upgraded their manuscript reporting their experience of CT-guided coil wire labeling for thoracoscopic resection of lung nodules in children and adolescents based on the reviewers’ comments. Minor revision on my comments below would be appreciated.

The authors described in Abstract “Our study shows that with a careful patient selection, thoracoscopic resection of lung metastases after coil wire labeling is a safe and reproducible procedure in children(probably “and adolescents”). It is described in Materials and Methods “Patients were selected for the presented procedure if indication for metastasis resection was foreseen by the protocol and if position of the lesions allowed minimally invasive resection but lesions were not visible on the lung surface”. The authors described in their response to reviewers “Other patients at our institution underwent different types of lung metastasis resection during the same period”.

I wonder what had been the indication for CT-guided coil wire labeling in the 13 years between 2010 and 2022 in their institutes, which is not described precisely and clearly in the text. I also wonder how they had managed such resections of lung nodules in children and adolescents, who had not been selected for CT-guided coil wire labeling before thoracoscopic resection, thoracotomy or thoracoscopic resections following fluorescing agents or radio-labeling for intraoperative detection of metastases.

I believe that concrete and specified descriptions regarding “a careful patient selection” for the CT-guided coil wire labeling in Abstract and Discussion would be informative and helpful for readers.

Authors reply: A detailed explanation about the selection criteria has now been added in the discussion section together with an introducing comment on the background of the selection using the presented method (page 7, 3rd paragraph). In order to keep the abstract compact and clearly arranged, the authors would prefer to add this detailed explanation only in the discussion. The authors thank the reviewer for his constructive and quality improving comments

Round 3
